# Evaluation of the Kinetics of Antibody Response to COVID-19 Vaccine in Solid Organ Transplant Recipients: The Prospective Multicenter ORCHESTRA Cohort

**DOI:** 10.3390/microorganisms10051021

**Published:** 2022-05-12

**Authors:** Maddalena Giannella, Elda Righi, Renato Pascale, Matteo Rinaldi, Natascia Caroccia, Chiara Gamberini, Zaira R. Palacios-Baena, Giulia Caponcello, Maria Cristina Morelli, Mariarosa Tamè, Marco Busutti, Giorgia Comai, Luciano Potena, Elena Salvaterra, Giuseppe Feltrin, Umberto Cillo, Gino Gerosa, Mara Cananzi, Salvatore Piano, Elisa Benetti, Patrizia Burra, Monica Loy, Lucrezia Furian, Gianluigi Zaza, Francesco Onorati, Amedeo Carraro, Fiorella Gastaldon, Maurizio Nordio, Samir Kumar-Singh, Mahsa Abedini, Paolo Boffetta, Jesús Rodríguez-Baño, Tiziana Lazzarotto, Pierluigi Viale, Evelina Tacconelli

**Affiliations:** 1Infectious Diseases Unit, Department of Integrated Management of Infectious Risk, IRCCS Azienda Ospedaliero-Universitaria di Bologna, 40138 Bologna, Italy; maddalena.giannella@unibo.it (M.G.); matteo.rinaldi@aosp.bo.it (M.R.); natascia.caroccia@unibo.it (N.C.); pierluigi.viale@unibo.it (P.V.); 2Department of Medical and Surgical Sciences, University of Bologna, 40138 Bologna, Italy; mahsa.abedini@unibo.it (M.A.); paolo.boffetta@unibo.it (P.B.); 3Division of Infectious Diseases, Department of Diagnostics and Public Health, University of Verona, 37134 Verona, Italy; elda.righi@univr.it (E.R.); evelina.tacconelli@univr.it (E.T.); 4Microbiology Unit, IRCCS Policlinico Sant’Orsola, University of Bologna, 40138 Bologna, Italy; chiara.gamberini15@unibo.it (C.G.); tiziana.lazzarotto@unibo.it (T.L.); 5Infectious Diseases and Microbiology Unit, Hospital Universitario Virgen Macarena and Department of Medicine, University of Sevilla/Biomedicines Institute of Sevilla, CSIC, 41009 Sevilla, Spain; zaira.palacios.baena@hotmail.com (Z.R.P.-B.); gcaponcello@gmail.com (G.C.); jesusrb@us.es (J.R.-B.); 6Centro de Investigación Biomédica en Red en Enfermedades Infecciosas (CIBERINFEC), 28029 Madrid, Spain; 7Internal Medicine Unit for the Treatment of Severe Organ Failure, IRCCS Azienda Ospedaliero-Universitaria di Bologna, 40138 Bologna, Italy; mariacristina.morelli@aosp.bo.it; 8Gastroenterology Unit, Department of Digestive, Hepatic and Endocrine-Metabolic Diseases, IRCCS Azienda Ospedaliero-Universitaria di Bologna, 40138 Bologna, Italy; mariarosa.tame@aosp.bo.it; 9Nephrology, Dialysis and Transplantation Unit, Department of Experimental, Diagnostic and Specialty Medicine, IRCCS Azienda Ospedaliero-Universitaria di Bologna, 40138 Bologna, Italy; marco.busutti@live.com (M.B.); giorgia.comai@aosp.bo.it (G.C.); 10Heart Failure and Transplant Unit, IRCCS Azienda Ospedaliero-Universitaria di Bologna, 40138 Bologna, Italy; luciano.potena2@unibo.it; 11Division of Interventional Pulmonology Unit, IRCCS Azienda Ospedaliero-Universitaria di Bologna, 40138 Bologna, Italy; elena.salvaterra@aosp.bo.it; 12Regional Center for Transplant Coordination, 35128 Padua, Italy; giuseppe.feltrin@regione.veneto.it; 13Hepatobiliary Surgery and Liver Transplantation Unit, Department of Surgery, Oncology and Gastroenterology, Padua University Hospital, 35128 Padua, Italy; cillo@unipd.it; 14Cardiac Surgery Unit, Department of Cardiac, Thoracic, Vascular Sciences and Public Health, University of Padua, 35128 Padua, Italy; gino.gerosa@unipd.it; 15Unit of Pediatric Gastroenterology, Digestive Endoscopy, Hepatology and Care of the Child with Liver Transplantation, Department of Women’s and Children’s Health, University Hospital of Padua, 35128 Padua, Italy; mara.cananzi@aopd.veneto.it; 16Unit of Internal Medicine and Hepatology (UIMH), Department of Medicine—DIMED, University of Padua, 35128 Padua, Italy; salvatorepiano@gmail.com; 17Pediatric Nephrology, Dialysis and Transplant Unit, Department of Women’s and Children’s Health, Padua University Hospital, 35128 Padua, Italy; elisa.benetti@aopd.veneto.it; 18Unit of Gastroenterology and Multivisceral Transplant, Department of Surgery, Oncology and Gastroenterology, University Hospital of Padua, 35128 Padua, Italy; burra@unipd.it; 19Thoracic Surgical Unit, Department of Cardiac, Thoracic, and Vascular Sciences, University of Padua, 35128 Padua, Italy; monica.loy@aopd.veneto.it; 20Kidney and Pancreas Transplantation Unit, Department of Surgical, Oncological and Gastroenterological Sciences, University of Padua, 35128 Padua, Italy; lucrezia.furian@unipd.it; 21Renal Unit, Department of Medicine, University Hospital of Verona, 37134 Verona, Italy; gianluigi.zaza@univr.it; 22Division of Cardiac Surgery, University of Verona, 37134 Verona, Italy; francesco.onorati@aovr.veneto.it; 23Liver Transplant Unit, Department of Surgery and Dentistry, University and Hospital Trust of Verona, 37134 Verona, Italy; amedeo.carraro@aovr.veneto.it; 24Department of Nephrology, Dialysis and Transplantation, San Bortolo Hospital, 36100 Vicenza, Italy; fiorella.gastaldon@aulss8.veneto.it; 25Nephrology, Dialysis and Transplantation Unit, Treviso Hospital, 35121 Treviso, Italy; maurizio.nordio@aulss2.veneto.it; 26Molecular Pathology Group, Laboratory of Cell Biology & Histology University of Antwerp, Faculty of Medicine, 2610 Antwerp, Belgium; samir.kumarsingh@uantwerpen.be; 27Stony Brook Cancer Center, Stony Brook University, Stony Brook, NY 11794, USA

**Keywords:** COVID-19, mRNA vaccines, serology, antibody response, solid organ transplantation

## Abstract

Previous studies assessing the antibody response (AbR) to mRNA COVID-19 vaccines in solid organ transplant (SOT) recipients are limited by short follow-up, hampering the analysis of AbR kinetics. We present the ORCHESTRA SOT recipients cohort assessed for AbR at first dose (t0), second dose (t1), and within 3 ± 1 month (t2) after the first dose. We analyzed 1062 SOT patients (kidney, 63.7%; liver, 17.4%; heart, 16.7%; and lung, 2.5%) and 5045 health care workers (HCWs). The AbR rates in the SOTs and HCWs were 52.3% and 99.4%. The antibody levels were significantly higher in the HCWs than in the SOTs (*p* < 0.001). The kinetics showed an increase (*p* < 0.001) in antibody levels up to 76 days and a non-significant decrease after 118 days in the SOT recipients versus a decrease up to 76 days (*p* = 0.02) and a less pronounced decrease between 76 and 118 days (*p* = 0.04) in the HCWs. Upon multivariable analysis, liver transplant, ≥3 years from SOT, mRNA-1273, azathioprine, and longer time from t0 were associated with a positive AbR at t2. Older age, other comorbidities, mycophenolate, steroids, and impaired graft function were associated with lower AbR probability. Our results may be useful to optimize strategies of immune monitoring after COVID-19 vaccination and indications regarding timing for booster dosages calibrated on SOT patients’ characteristics.

## 1. Introduction

COVID-19 (coronavirus disease 2019) is caused by the severe acute respiration syndrome coronavirus 2 (SARS-CoV-2). Due to the rapid spread and high morbidity and mortality burden of COVID-19, major efforts were undertaken regarding the development of vaccines using preexisting or novel technologies [1,2]. Overall, the mRNA vaccines showed higher rates of protection than other vaccine types against severe disease and mortality, as well as against new variants of concern, and possessed an optimal safety profile [2,3]. This type of vaccine contains a nucleoside-modified mRNA that encodes the SARS-CoV-2 spike glycoprotein, eliciting both B- and T-cell responses and inducing a prolonged antibody production [2]. Due to their efficacy and safety profiles, mRNA vaccines have been considered the first choice for protecting immunocompromised patients, such as hematological patients and solid organ transplant recipients, by some health systems (https://www.aifa.gov.it/ (accessed on 17 April 2022). In addition, mRNA vaccines were approved and introduced before other vaccine types; thus, the majority of the early data on COVID-19 vaccination in prioritized categories were based on the use of mRNA vaccines.

Indeed, to date, several reports have underlined low rates of antibody response to mRNA COVID-19 vaccines in solid organ transplant (SOT) recipients [4,5,6,7,8,9,10,11,12,13,14,15,16,17,18,19,20,21,22,23,24,25]. The largest report is that from Boyarsky et al., including 658 recipients of different types of SOT (322 kidney, 129 liver, 97 heart, 71 lung, 22 multiorgan, and 5 pancreas) recruited across several US hospitals [5]. The antibody response after the first and second dose of an mRNA COVID-19 vaccine was assessed at the second dose and within 21 days thereafter, showing a response rate of 15% and 54%, respectively [5]. The major flaws of previous studies were a small sample size, a lack of a control group, and a limited follow-up period, hampering the analysis of the antibody response. The only study assessing the kinetics of the antibodies is a sub-study of 305 SOT recipients from the Boyarsky cohort, where the patients were sampled at three timepoints (before the second dose, 1 month, and 3 months after the second dose of mRNA vaccine). The study showed that the antibody response was largely stable after 3 months following vaccination [26]. The understanding of the kinetics of the antibody response over time in SOT recipients can play a pivotal role both for public health officials when developing recommendations for vaccination needs and booster schedules and for physicians in terms of optimizing monitoring studies and designing tailored preventive strategies. The aim of this study was to analyze, in a large multicenter cohort of SOT recipients, the kinetics of the serological response over time compared with a cohort of healthcare workers (HCWs) and drivers of the immune response to two doses of mRNA COVID-19 vaccines.

## 2. Materials and Methods

The study is part (Workpackage 4) of the Horizon2020 ORCHESTRA project (https://orchestra-cohort.eu/ (accessed on 17 April 2022), which aims to create a new pan-European cohort to rapidly advance the knowledge on the COVID-19 infection. The project currently includes multiple cohorts, e.g., individuals at risk of infection, COVID-19 patients, HCWs, and fragile populations, such as SOT recipients, reaching a current total of more than 1,300,000 subjects at the time of writing of this manuscript.

The prospective observational multicenter cohort study of SOT recipients includes all consecutive adult (≥18 years) SOT patients who received two doses of mRNA COVID-19 vaccine, according to national health system guidance documents, between January and May 2021, and it is running at six hospitals, five from Italy (Bologna, Verona, Padova, Vicenza, and Treviso) and one from Spain (Seville). For the purpose of this analysis, patients with clinical and/or immunological evidence of prior COVID-19 were excluded. As control group, a cohort of 5045 HCWs from the Bologna ORCHESTRA cohort (Workpackage 5) vaccinated with two doses of mRNA COVID-19 vaccine in the same study period and without history of SARS-CoV-2 infection were analyzed. Primary endpoints were the probability of positive AbR at 3 ± 1 month of first vaccine dosage in SOT recipients and the kinetics of AbR in SOT recipients compared with HCWs. A positive AbR was defined as an anti-rapid binding domain (RBD) titer ≥5 U/mL or ≥45 BAU/mL for the Elecsys and MSD assays, respectively (see following serology section for details). Secondary endpoints included the clinical and epidemiological drivers of positive AbR in SOT recipients. According to the study protocol, all SOT recipients had serological response assessed at the following timepoints: the day of first vaccine dose administration (t_0_); the day of second dose administration (t_1,_ 21 or 28 days after t_0_ for BNT162b2 or mRNA-1273, respectively); and 3 ± 1 month after t_0_ (t_2_) (see Appendix A).

Data were collected at t_0_ and included: age, sex, comorbidities other than the cause of transplant according to Charlson index criteria, type and date of transplant, current immunosuppressive regimen, receipt of induction regimen in the last 6 months, and graft function defined as good, impaired, or failure according to the judgement of attending physicians. Occurrence of SARS-CoV-2 infection and clinical course was collected at each timepoint. Data collected for the HCWs at the same timepoint included age, sex, and AbR. Study variables were registered using a standardized electronic case report form (eCRF) managed by a centralized REDCap capture tool [27]. Data sources were clinical charts and hospital electronic records. Detection of AbR was performed at Bologna University, Italy (for the Bologna cohort of SOT recipients and HCWs), and at Antwerp University, Belgium (for all the other cohorts) with Elecsys^®^ Anti-SARS-CoV-2 ECLIA assay (Roche Diagnostics AG, Rotkreuz, Switzerland) and V-PLEX SARS-CoV-2 Panel 6 Kit (IgG) from Meso Scale Discovery (MSD, Rockville, MD, USA), respectively. The Elecsys^®^ Anti-SARS-CoV-2 ECLIA assay (Roche Diagnostics AG, Rotkreuz, Switzerland) was performed on the Cobas e 801 analyzer (Roche Diagnostics). The cut-off value for positive reactivity anti-N was equal to 1.0 COI (cut-off index), and anti-S (RBD) was 0.8 U/mL, according to the manufacturer’s instructions. To establish more accurate criteria for interpretation of serological results, the assay was validated in two well-defined groups of serum samples obtained from 50 HCWs before and one month after receiving the second dose of vaccine. Based on the results from the 50 true-positive and the 50 true-negative SARS-CoV-2 samples, antibody responses were stratified according to Anti-N (negative: 1.0 COI; inconclusive: ≥1 to <5 COI; positive: ≥5 COI) and Anti-S (negative: 0.8 U/mL; inconclusive: ≥0.8 to <5 U/mL; positive: ≥5 U/mL). The V-PLEX SARS-CoV-2 Panel was used according to the manufacturer’s instructions. IgG titers to the following antigens were measured: SARS-CoV-2 N, SARS-CoV-2 S1 RBD, SARS-CoV-2 Spike, SARS-CoV-2 Spike (D614G), SARS-CoV-2 Spike (B.1.1.7), SARS-CoV-2 Spike (B.1.351), SARS-CoV-2 Spike (P.1). Quantitative IgG results were measured in antibody units (AU)/mL, converted to WHO binding antibody units (BAU)/mL using a conversion factor provided by MSD. The detection range is described in Appendix A. The overall antibody responses were stratified into non-reactive, inconclusive, positive-low, positive-mild, and positive-high according to WHO criteria (see Appendix A).

## 3. Results

By November 2021, 1452 SOT recipients were included in the ORCHESTRA SOT cohort. From this number, 412 were excluded from this analysis for the following reasons: the results of the AbR at 3 ± 1 month were not available at the time of analysis due to logistical issues (*n* = 304), SARS-CoV-2 infection before vaccination (*n* = 76) or between dosages (*n* = 8), and incomplete vaccination schedule (*n* = 23). The eight breakthrough infections occurred in seven kidney and one liver transplant recipients within a mean of 16.62 days (range 8–33) from the first dose administration of BNT162b2 and mRNA-1273 vaccine in seven and one SOT recipient, respectively.

Therefore, a total of 1062 patients were included in the analysis. The majority of the enrolled patients were males (704, 66.3%), with a mean age (±SD) of 58.28 (±13.10) years. Concomitant comorbidities (other than the cause of SOT) were present in 715 (67.3%) patients. As for the type of SOT, most patients had kidney (*n* = 677, 63.7%), followed by liver (*n* = 182, 17.4%), heart (*n* = 111, 16.7%), and lung (*n* = 26, 2.5%) transplantation. In the majority of the patients (*n* = 836, 78.7%), more than 3 years had elapsed from SOT to administration of vaccination. Accordingly, only one patient had received an induction regimen in the last 6 months prior to vaccination. The most common drugs used for maintenance immunosuppressive regimen were tacrolimus (*n* = 763, 72%), steroids (*n* = 709, 66.7%), and mycophenolate mofetil (*n* = 626, 59%). Overall, 222 (20.9%) patients were reported as having an impaired function of the graft or graft failure. BNT162b2 andmRNA-1273 vaccines were administered to 928 (87.4%) and 134 (12.6%) patients, respectively. The cohort details are described in Table 1. The number of SOT recipients enrolled at each center and the distribution of the types of graft per center are shown in Appendix A.

All the SOT recipients tested at t_0_ had a negative AbR (*n* = 622). The rate of positive AbR was 9.8% (62/631) at t_1_ and 52.3% (556/1062) at t_2_. The mean time from the first vaccination to t_2_ was 92 ± 35 days, with the majority of the patients being assessed between 70 and 100 (*n* = 500, 47.08%) and 100 and 130 (*n* = 298, 28.06%) days after the first dose (see Table 1). The analysis of the weekly trend of odds for positive AbR showed a steady increase in the probability of having a positive response from day 50 to day 110 after the first dose of vaccination. The ORCHESTRA HCWs cohort included 5045 subjects (68.9% women, mean age 43.1 years). The positions of the healthcare workers enrolled are summarized in Appendix A. A serological response was detected in 99.5% of the subjects at t_2_. Breakthrough infections were found in 32 (0.63%) HCWs between 40 and 150 days after the 1st dose of vaccine and in 73 (1.45%) HCWs after 14 days since full vaccination. In the multivariable logistic regression analysis, the adjusted OR of the serological response for HCWs versus SOT recipients was 120 (95% CI 73.1–199). The adjusted mean of ln (AbR) was 6.9 (±0.01) in the HCWs and 5.2 (±0.05) in SOT recipients (*p* < 0.001). Figure 1 shows the mean ln (AbR) in the two populations for the periods between 49 and 153 days (the overlap period of the two series of results) after vaccination in individuals with positive AbR. The ratio of ln (AbR) between HCWs and SOT recipients ranged between 1.2 and 1.7, i.e., between 3.3 and 5.5 on an arithmetic scale; the SOT recipients showed a significant increase up to 76 days (*p* < 0.001), then a non-significant decrease in ln (AbR) after 118 days (*p* = 0.1); conversely, the HCWs experienced a decrease in ln (AbR) up to 76 days (*p* = 0.02) and a less pronounced decrease between 76 and 118 days (*p* = 0.04).

To predict the drivers of AbR in SOT recipients, univariable and multivariable analyses were performed. Univariable analysis found significant differences between the patients with positive and negative AbR according to age, presence of other comorbidities, type of graft, time from SOT, immunosuppressive drugs (tacrolimus, mycophenolate mofetil, azathioprine, everolimus, and steroids), impaired graft function or graft failure, and the type of mRNA COVID-19 vaccine (see Table 1). Upon multivariable analysis, liver transplant (vs. other types of SOT; OR 2.71, 95%CI 1.55 4.72, *p* < 0.001), ≥3 years from SOT to vaccination (OR 4.92, 95%CI 2.56–9.45, *p* < 0.001), mRNA-1273 vaccine (3.57, 95%CI 2.25 5.67, *p* < 0.001), use of azathioprine (OR 3.43, 95%CI 1.20–9.82, *p* = 0.02), and longer time from vaccination to serological assessment (OR 1.30, 95%CI 1.10–1.53, *p* < 0.001) were associated with a positive AbR. Meanwhile, older age (OR 0.68, 95%CI 0.60–0.77, *p* <0.001), presence of other comorbidities (OR 0.60, 95%CI 0.43 0.83,*p* = 0.002), use of mycophenolate mofetil (OR 0.29, 95%CI 0.20 0.43, *p* < 0.001), steroids (OR 0.44, 95%CI 0.30–0.65, *p* < 0.001), and impaired graft function or graft failure at vaccination (OR 0.38, 95%CI 0.26–0.55, *p* < 0.001) were associated with a lower probability of positive antibody response at t_2_ (see Table 2).

### Statistical Analysis

For descriptive analysis, categorical variables were presented as absolute numbers, and their relative frequencies and continuous variables were presented as mean ± standard deviation (SD) if normally distributed, or as median and interquartile range (IQR) if non-normally distributed. Quantitative anti-RBD levels for positive cases were log-transformed to account for the skewness of the distribution and then normalized by dividing them by the center-specific standard error to take into account the different methods used across centers. The comparison between SOT recipients and HCWs was performed using multivariable logistic regression with AbR as dependent variable and cohort (HCWs vs. SOT recipients) as primary endpoint, after adjustment for sex, age, and time since vaccination. Sex- and age-adjusted means of anti-RBD levels were calculated among subjects with positive immune response in the two groups, based on ANOVA. Time trends in log-transformed anti-RBD levels were assessed with linear regression after application of linear splines with two knots at weeks 10 and 16. For the secondary endpoint, multivariable logistic regression models were fitted to estimate odds ratios (ORs) and 95% confidence intervals (CI) of AbR as a dichotomous variable. The main exposure variable was time between administration of first vaccine dose and the AbR assessment; models also included sex, age (categorical), comorbidities, type of graft, type of vaccine, time between transplant and vaccination (categorical), induction regimen in the last 6 months, immunosuppressive drugs at the time of vaccination (calcineurin inhibitors, anti-metabolites, mTOR inhibitors, steroids), graft function (good, impaired, failure), and time between first dose and assessment of AbR (categorical) as potential confounders. Analyses were completed using the STATA package, Version 16.1 (STATA/SE 16.0 for Windows. StataCorp Llc., College Station, TX, USA) using the commands *logistic*, *glm*, *anova*, and *mksplines*.

The study, according to the Italian legislation for SARS-CoV-2 studies, was approved by the Agenzia Italiana del Farmaco (AIFA) and the Ethics Committee of Istituto Nazionale per le Malattie Infettive (INMI) Lazzaro Spallanzani (document n. 359 of Study’s Registry 2020/2021). Informed consent was obtained from all the enrolled patients. The study was conducted in accordance with the Declaration of Helsinki.

## 4. Discussion

The ORCHESTRA SOT cohort is the largest cohort of SOT recipients assessed for a serological response to mRNA SARS-CoV-2 vaccines reported to date and the first providing data regarding the kinetics of the antibody response compared to that of HCWs. The assessment of the immune response at month 3 ± 1 after two doses of mRNA COVID-19 vaccine found that 52.3% among 1062 naïve SOT recipients had a positive antibodies response versus 99.5% detected in HCWs. A steady increase in the probability of having a positive response from day 50 to day 110 after first dose administration was observed. Furthermore, as expected among the serological responders, the mean levels of antibodies were significantly higher in the HCWs than in the SOT recipients, and the kinetics was different. Our results suggest that early assessment of AbR in SOT recipients could miss the subsequent increase in AbR, and, on the other hand, that SOT recipients maintain good antibody levels during a limited period of time (up to day 118) after a standard vaccination schedule (two doses of mRNA COVID-19 vaccine).

The rate of seroconversion was heterogeneous by type of SOT, as already observed [28], and varied from 46% in kidney- to 79% in liver transplant recipients. It is noteworthy that factors such as age, comorbidities, type of graft, time from SOT, graft function, and type of immunosuppressive regimen were associated with the AbR within 3 months. Therefore, prevention strategies, other than vaccination, could be considered in this population, such as reduction in immunosuppression [29] or the use of long-acting monoclonal antibodies against the spike protein of SARS-CoV-2 [30].

A higher intensity of immunosuppressive regimen, in particular the use of anti-metabolites drugs, has been associated with a lower antibody response [28]. For this reason, some authors have proposed the temporary suspension of mycophenolate during vaccine administration, although this practice has been discouraged by international transplant societies due to safety concerns and a lack of data regarding its efficacy (https://ishlt.org/ishlt/media/documents/ISHLT-AST_SARS-CoV-2-Vaccination_5-11-21.pdf (accessed on 17 April 2022)). A clinical trial is currently ongoing assessing the seroconversion rate after a third dose among patients receiving an mRNA vaccination with or without temporary suspension of mycophenolate [31]. We confirmed the negative role of mycophenolate, along with steroids, on the probability to achieve a seroconversion rate at 3 ± 1 month from vaccination, while patients on azathioprine were more likely to show a positive AbR. We deem that this result could be relevant for future strategies to improve the efficacy of vaccination in SOT recipients, considering azathioprine as a temporary alternative to mycophenolate, in order to improve the immunological response and minimize the rejection risk at the same time.

Data on the kinetics of the antibody response to SARS-CoV2 infection or vaccine, as well as to other infectious agents, such as influenza, are limited [32]. Most studies focused on the duration of the antibody response, with the waning of antibodies being the main concern of physicians and public health officials during the pandemic of SARS-CoV-2. However, the knowledge of the time needed to mount a protective AbR in SOT recipients could be very important to provide correct advice to patients and plan adequate monitoring activities. Our study shows a steady increase in the probability of having a positive antibody response between day 50 and 110 after the first vaccination, suggesting that this could be the best time interval to assess the antibody response to vaccination in SOT recipients. Similarly, Bovarsky et al. observed an increase from 13.5% to 67% at month 3 after the second dose [26], confirming that a delayed sampling could be associated with a higher probability of finding a positive antibody response to a COVID-19 vaccine in SOT recipients. The use of immunosuppressive medications that inhibit T-cell and B-cell responses to prevent transplant rejection could have a role in this delayed response [33]. Very interestingly, our data also showed a different kinetics of the antibody response in SOT recipients compared to HCWs, which merits further investigation. Our data confirmed a lower level of antibodies among SOT recipients than HCW responders, which was maintained during a limited period of time, supporting the current strategy of booster dosage in this setting. In this regard, the preliminary data were controversial, showing a moderate rate of seroconversion [34,35]. According to our and prior data, low-level responders could be those who most benefit from booster dosages [33].

A limitation of our study is the lack of cellular immune response analysis. Indeed, some authors have shown, upon analyzing cellular immunity as well, that the rate of overall immunological response seems to be higher than that reported considering only the serological response, mainly among patients receiving hybrid (vector/mRNA) COVID-19 vaccination [8,14,36]. Furthermore, the association between the rates and patterns of the immunological response and clinical effectiveness of COVID-19 vaccination in SOT recipients is still an open issue that should be investigated in future studies. We did not collect the dosage of immunosuppressive drugs at vaccination, which could impact the AbR to vaccines in SOT recipients. Finally, due to the high volume of patients and in order to avoid multiple visits to SOT recipients with already scheduled visits near the predefined timepoint, we left a wider temporal range to local sites for performing the 3M visits; thus, 30% of the visits fell out of the predefined interval. However, we deem that the wide temporal range could have allowed us to capture differences in the kinetic of antibody response between SOT recipients and HCWs. The strengths of our study, in addition to the large study population, include the extensive amount of clinical data available for analysis, enabling a detailed investigation of the characteristics of the SOT recipients associated with an immune response to mRNA vaccines, the fact that a 3-month follow-up was available for all the subjects, and the use of a comparison population of HCWs.

## 5. Conclusions

In conclusion, we showed that a positive antibodies response at month 3 ± 1 after two doses of mRNA COVID-19 vaccine was found in 52.3% of the SOT recipients versus 99.5% in HCWs. However, the timing of the sampling was significantly associated with the probability of finding a positive antibody response to the COVID-19 vaccine in the SOT recipients, suggesting a slow increase in the antibody levels. Indeed, when compared with the HCWs, the kinetics of the antibody response in the SOT recipients was different and was maintained at good levels during a limited period of time. These results could be helpful to optimize strategies of immune monitoring after COVID-19 vaccination and to provide indications regarding the timing for booster dosages in the setting of SOT recipients.

## Figures and Tables

**Figure 1 microorganisms-10-01021-f001:**
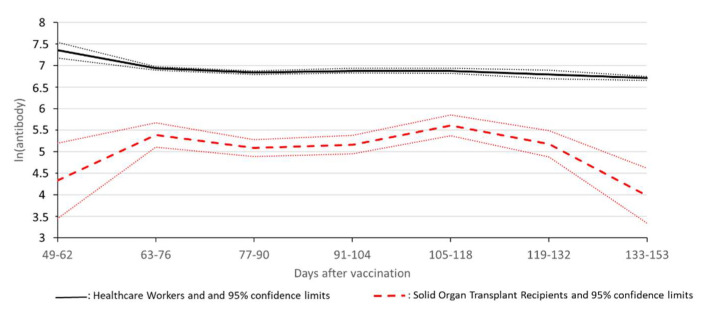
Mean ln (RBD) and 95% confidence limits in HCWs (continuous line) and SOT recipients (broken line) between 49 and 153 days after vaccination, adjusted for sex and age. The mean ln (AbR) in the two populations for the periods between 49 and 153 days after vaccination in individuals with positive AbR. The ratio of ln (AbR) between HCWs and SOT recipients ranged between 1.2 and 1.7, i.e., between 3.3 and 5.5 on arithmetic scale; SOT recipients showed a significant increase up to 76 days (*p* < 0.001), then a non-significant decrease in ln (AbR) after 118 days (*p* = 0.1); conversely, HCWs experienced a strong decrease in ln (AbR) up to 76 days (*p* = 0.02), and a less pronounced decrease between 76 and 118 days (*p* = 0.04).

**Table 1 microorganisms-10-01021-t001:** General characteristics of study population.

	Total *n* = 1062 (%)	Positive Antibody Response at 3 ± 1 Month N = 556 (%)	Negative Antibody Response at 3 ± 1 Month N = 506 (%)	*p*
**Demographic data**				
Age (mean ± SD) (years)	58.28 ± 13.10	56.47 ± 13.61	60.26 ± 12.23	<0.001
Age group				<0.001
<39 y	93 (8.76%)	63 (67.74%)	30 (32.26%)	
40–49 y	156 (14.69%)	93 (59.62%)	63 (40.38%)	
50–59 y	268 (25.24%)	146 (54.48%)	122 (45.52%)	
60–69 y	309 (29.10%)	148 (47.90%)	161 (52.10%)	
≥70 y	236 (22.22%)	106 (44.92%)	130 (55.08%)	
Sex				0.967
Male	704 (66.29%)	368 (52.27%)	336 (47.73%)	
Female	353 (33.40%)	185 (52.41%)	168 (47.59%)	
**Comorbidities**				0.005
No	347 (32.67%)	203 (58.50%)	144 (41.5%)	
Yes	715 (67.33%)	353 (49.37%)	362 (50.63%)	
**Type of graft**				<0.001
Kidney	677 (63.75%)	312 (46.09%)	365 (53.91%)	
Heart	177 (16.67%)	86 (48.59%)	91 (51.41%)	
Liver	182 (17.14%)	144 (79.12%)	38 (20.88%)	
Lung	26 (2.45%)	14 (53.85%)	12 (46.15%)	
**Type of vaccine**				<0.001
BNT162b2	928 (87.38%)	463 (49.89%)	465 (50.11%)	
mRNA-1273	134 (12.62%)	93 (69.40%)	41 (30.60%)	
**Time from transplant to vaccination**				<0.001
Less than 1 year	58 (5.46%)	21 (36.21%)	37 (63.79%)	
1 to 3 years	166 (15.63%)	68 (40.96%)	98 (59.04%)	
More than 3 years	836 (78.72%)	465 (55.62%)	371 (44.38%)	
**Induction regimen in the last 6 months**				0.340
No	1061 (99.90%)	555 (52.31%)	506 (47.69%)	
Any	1 (0.09%)	1 (100%)	0 (0.00%)	
**Immunosuppressive drugs at the time of vaccination**				
Calcineurin inhibitors	1007 (94.82%)	520 (51.64%)	487 (48.36%)	0.046
Tacrolimus	763 (71.85%)	384 (50.33%)	379 (49.67%)	0.035
Cyclosporine	246 (23.16%)	136 (55.28%)	110 (44.72%)	0.294
Anti-metabolites	663 (62.43%)	284 (42.84%)	379 (57.16%)	<0.001
Mycophenolate mofetil	626 (58.95%)	252 (40.26%)	374 (59.74%)	<0.001
Azathioprine	37 (3.48%)	32 (86.49%)	5 (13.51%)	<0.001
mTOR	144 (13.56%)	90 (62.50%)	54 (37.50%)	0.009
Everolimus	128 (12.05%)	78 (60.94%)	50 (39.06%)	0.038
Sirolimus	16 (1.51%)	12 (75.00%)	4 (25.00%)	0.068
Steroids	709 (66.76%)	312 (44.01%)	397 (55.99%)	<0.001
**Impaired graft function**				<0.001
Good	830 (78.15%)	475 (57.23%)	355 (42.77%)	
Impaired or Failure	222 (20.90%)	75 (33.78%)	147 (66.22%)	
**Time between first dose and assessment of antibody response**				0.038
40–70 d	170 (16.01%)	74 (43.53%)	96 (56.47%)	
70–100 d	500 (47.08%)	262 (52.40%)	238 (47.60%)	
100–130 d	298 (28.06%)	170 (57.05%)	128 (42.95%)	
130–160 d	75 (7.06%)	37 (49.33%)	38 (50.67%)	
>160 d	19 (1.79%)	13 (68.42%)	6 (31.58%)	

mTOR: mammalian target of rapamycin; SD: standard deviation.

**Table 2 microorganisms-10-01021-t002:** Multivariable analysis of predictors of antibody response at 3 ± 1 months after first dose administration of mRNA COVID-19 vaccine in SOT recipients.

Variable	OR (95% CI)	*p*-Value (α = 0.05)
**Sex**	-	-
Male	1 (ref)	-
Female	0.91 (0.67 1.24)	0.568
**Age**	-	-
Categorical increase (<39 y; 40–49 y; 50–59 y; 60–69 y; ≥70 y)	0.67 (0.60 0.76)	<0.001
**Type of graft**	-	-
Kidney	1 (ref)	
Heart	0.64 (0.39 1.07)	0.090
Liver	2.71 (1.55 4.72)	<0.001
Lung	1.16 (0.46 2.95)	0.750
**Time from transplant to vaccination**	-	-
Less than 1 year	1 (ref)	
1 to 3 years	1.79 (0.87 3.67)	0.111
More than 3 years	4.92 (2.56 9.45)	0.000
**Time from vaccination onset to serological assessment**		
Categorical increase (40–70 d; 70–100 d; 100–130 d; 130–160 d; >160 d)	1.30 (1.10 1.53)	<0.001
**Comorbidities**		
No	1 (ref)	
Yes	0.60 (0.43 0.83)	0.002
**Type of vaccine**		
BNT162b2	1 (ref)	
mRNA-1273	3.57 (2.25 5.67)	<0.001
**Immunosuppressive drugs at the time of vaccination**		
Cyclosporine	0.71 (0.30 1.67)	0.429
Tacrolimus	0.52 (0.23 1.16)	0.111
Azathioprine	3.43 (1.20 9.82)	0.022
Mycophenolates	0.29 (0.20 0.43)	<0.001
Sirolimus	0.70 (0.18 2.66)	0.598
Everolimus	0.72 (0.43 1.20)	0.212
Steroids	0.44 (0.30 0.65)	0.000
**Impaired graft function**		
Good	1 (ref)	
Impaired, Failure, and others	0.38 (0.26 0.55)	<0.001

OR: odd ratio.

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
