# Peer review of "Evaluation of the Kinetics of Antibody Response to COVID-19 Vaccine in Solid Organ Transplant Recipients: The Prospective Multicenter ORCHESTRA Cohort"

_microorganisms, 2022, doi:10.3390/microorganisms10051021_

Round 1
Reviewer 1 Report
The article by Maddalena Giannella et colleagues provides useful information about the evaluation of the kinetics of antibody response to the COVID-19 vaccine in solid organ transplant recipients.
The manuscript is interesting, but there are still some points to be addressed by the authors in order to improve their manuscript.
Comments and suggestions:
1. A scheme with all the steps of this study should be included
- Introduction section: The authors go directly into the subject of the paper without making a small background of SARS-CoV-2 infection, its mutations, vaccination and post-vaccination antibodies. These things must be added to make a connection with the purpose of the work. Revise it. Line 76: Which type of COVID-19 vaccine? Line 94: Justify the rationale for choosing this type of COVID-19 vaccine for the study.
- Figure 1. It is difficult to understand the meaning because the colors are black. I suggest a clearer figure with different colored lines. It is also necessary to add a legend with what these continuous lines represent and those two broken lines.
- Line 257: mention the type of COVID-19 vaccine.
5.Lines: 267-269: What does it mean :”a limited period of time after a standard vaccination schedule”? Mention this period of time
- Lines 272-275: the authors state: “…suggesting that prevention strategies, other 274 than vaccination, could be considered in this population-based on patient-specific characteristics”. Explain other strategies to prevent SARS-CoV-2 infection. Argue with scientific evidence/data on how mRNA vaccines prevent SARS-CoV-2 infection.
- Lines 277-279: Add a few examples regarding safety concerns and lack of data about mRNA vaccines efficacy.
- Lines 296-299: Here, the authors compare the results of their study obtained between 50 and 110 days after the first dose of vaccinations; with the results of another study by Bovarsky et al. obtained three months after the second vaccination dose. Why? These studies are very different. Justify and clarify this statement.
9. Lines 300-306: Mention more clearly what is already known on this topic and what this study adds. What perspectives for humans does this MS have?
- The conclusions are not supported by robust data. Revise it
Consider revision accordingly.
Reviewer 2 Report
This is an important and very well documented prospective observational study.
Could you please make some small clarifications?
“vaccinated with two doses of mRNA COVID-19 109 vaccine in the same study period and without history of SARS-CoV-2 vaccination were 110 analysed.”
[Do you mean without prior history of vaccination before the two doses?}
“As control group, a cohort of 5045 HCWs from the Bologna 108 ORCHESTRA cohort (Workpackage 5) vaccinated with two doses of mRNA COVID-19”
[can you please provide more data about the HCWs and their potential exposure to COVID-19compared to the SOT patients? e.g how many were in administrative positions, surgeons, medical clinicians, home visit nurses, in primary care clinics, in hospitals, In ICUs?]
“3±1 month of first vaccine dosage in SOT [I needed to read ahead to your table to find that the range was 40 160+ days; 70% of your sample were followed from 70 to 130 days]
“HCWs experienced a strong decrease in ln(AbR) up to 76 days (p=0.02), and a less pronounced decrease between 76 and 118 days (p=0.04).” [do you really think the decrease is large, inspecting the numbers on the vertical axis of the graph? clinically significant?]
Did any HCWs acquire a COVID infection after vaccination?
Were the 304 patients whose data were missing differ in characteristics potentially related to the outcome compared to those who remained in the trial? If so you could comment that an ITT analysis might or might not be significantly different from the per protocol analysis.
Round 2
Reviewer 1 Report
No answer given.